# Metabolomic Analysis of Actinic Keratosis and SCC Suggests a Grade-Independent Model of Squamous Cancerization

**DOI:** 10.3390/cancers13215560

**Published:** 2021-11-05

**Authors:** Valeria Righi, Camilla Reggiani, Elisabetta Tarentini, Adele Mucci, Alessia Paganelli, Anna Maria Cesinaro, Ema Mataca, Shaniko Kaleci, Barbara Ferrari, Marco Meleti, Cristina Magnoni

**Affiliations:** 1Department for Life Quality Studies, University of Bologna, 47921 Rimini, Italy; valeria.righi2@unibo.it; 2Dermatology Unit, Surgical, Medical and Dental Department of Morphological Sciences Related to Transplant, Oncology and Regenerative Medicine, University of Modena and Reggio Emilia, 41125 Modena, Italy; camilla.reggiani@unimore.it (C.R.); elisabetta.tarentini@unimore.it (E.T.); alessia.paganelli@unimore.it (A.P.); shaniko.kaleci@unimore.it (S.K.); ferrari.barbara@aou.mo.it (B.F.); 3Clinical and Experimental Medicine Ph.D. Program, University of Modena and Reggio Emilia, 41125 Modena, Italy; 4Department of Chemical and Geological Sciences, University of Modena and Reggio Emilia, 41125 Modena, Italy; adele.mucci@unimore.it; 5Department of Anatomic Pathology, University Hospital of Modena, 41100 Modena, Italy; cesinaro.annamaria@aou.mo.it (A.M.C.); ema.mataca@auslromagna.it (E.M.); 6Center of Dental Medicine, Department of Medicine and Surgery, University of Parma, 43126 Parma, Italy; marco.meleti@unipr.it

**Keywords:** human skin metabolomics, actinic keratosis, squamous cell carcinoma, metabolomic NMR profiling, histology, biomarkers

## Abstract

**Simple Summary:**

Actinic keratoses (AKs) are the most common sun-induced precancerous lesions that can progress to squamocellular carcinoma (SCC). AK I have been considered low-risk lesions, often evolving into AK II, the AK grade II and III have the potential to evolve to SCC. This research has assessed the metabolomic fingerprints of AK I, AK II, AK III and SCC by HR-MAS NMR spectroscopy, with the aim of evaluating the hypothesis of grade-association AK to SCC. The association between AKs and SCCs has also been evaluated by histopathology. Our findings support the notion that AK I are different from healthy skin and share different features with SCCs, indeed, they are metabolically active lesions with metabolic profiles similar to high-grade AKs and to SCC. The negative association of AKs with parakeratosis and the positive association with hypertrophy also suggested a similar behavior between AKs and SCCs. Therefore, all AKs should be treated independently from their clinical appearance or histological grade, since it is not possible to predict their potential evolution to SCC.

**Abstract:**

Background—Actinic keratoses (AKs) are the most common sun-induced precancerous lesions that can progress to squamocellular carcinoma (SCC). Recently, the grade-independent association between AKs and SCC has been suggested; however, the molecular bases of this potential association have not been investigated. This study has assessed the metabolomic fingerprint of AK I, AK II, AK III and SCC using high resolution magic angle spinning (HR-MAS) nuclear magnetic resonance (NMR) spectroscopy in order to evaluate the hypothesis of grade-independent association between AK and SCC. Association between AKs and SCCs has also been evaluated by histopathology. Methods—Metabolomic data were obtained through HR-MAS NMR spectroscopy. The whole spectral profiles were analyzed through multivariate statistical analysis using MetaboAnalyst 5.0. Histologic examination was performed on sections stained with hematoxylin and eosin; statistical analysis was performed using STATA software version 14. Results—A group of 35 patients affected by AKs and/or SCCs and 10 healthy controls were enrolled for metabolomics analysis. Histopathological analysis was conducted on 170 specimens of SCCs and AKs (including the ones that underwent metabolomic analysis). SCCs and AK I were found to be significantly associated in terms of the content of some metabolites. Moreover, in the logistic regression model, the presence of parakeratosis in AKs appeared to be less frequently associated with SCCs, while AKs with hypertrophy had a two-fold higher risk of being associated with SCC. Conclusions—Our findings, derived from metabolomics and histopathological data, support the notion that AK I are different from healthy skin and share some different features with SCCs. This may further support the expanding notion that all AKs should be treated independently from their clinical appearance or histological grade because they may be associated with SCC.

## 1. Introduction

Cutaneous squamous cell carcinoma (SCC) is among the most common skin cancers, second only to basal cell carcinoma [1,2,3]. Its incidence has increased over the last two decades leading to significant morbidity, mortality and health-related costs [4,5].

Actinic keratoses (AKs) are the most common sun-induced precancerous lesions that can progress to SCC. The actual risk of progression from AK to SCC is unknown and has been differentially estimated, with progression rates ranging from 0.025% to 20% per year [6,7,8].

Historically, the progression from AK to SCC was proposed to occur according to a multi-step pathway, similar to that of evolving carcinoma of the uterine cervix [9]. In this pathway, AK grade I (AK I), characterized by atypical keratinocytes localized in the basal cell layer of epidermis, would progress to AK grade II (AK II), in which cell atypia involves basal, granular and spinous layers and eventually AK grade III (AK III), identified by full thickness epidermal atypia [10]. According to this model, grade II and III lesions have the potential to evolve to SCC.

On the other hand, AK I have been considered as low-risk lesions, often evolving into AK II or rarely regressing. However, a recent study showed that AK I with atypical cells in the basal layer are frequently associated with invasive SCC [11]. The “differentiated pathway” does not require a full thickness atypia for malignant transformation: SCC develops directly from an epidermis showing alterations only in the basal layer [11]. Therefore, the association between AK and SCC may exist regardless of AK grade potentially suggesting a grade-independent progression to SCC [11]. However, the molecular bases of this potential association have not been investigated to date. Furthermore, it was recently proposed that AKs can be considered as an intraepithelial neoplasia per se, regardless of their potential progression to SCC, since dysplastic keratinocytes of AK present features similar to those of invasive SCC [12,13].

Metabolomics is the global and systematic assessment of metabolites in a biological system, capturing the metabolic perturbations driving physiological and disease statuses. Advancement of analytical techniques such as nuclear magnetic resonance (NMR) has enabled the quantitative identification of a wide range of metabolites using a very small volume of sample [14]. Metabolic profiling was already applied to the study of cancer biology [15,16,17,18], also with the aim of identifying potential predictive disease biomarkers. A recent study already identified potential biomarkers of AK through integrated metabolomics and histopathological analysis, with the aim of investigating the effects of the field cancerization treatment [19]. However, metabolomics has been scantly applied to skin cancers and the association between AK lesions and SCC.

This study assessed the metabolomic fingerprints of AK I, AK II, AK III and SCC using HR-MAS NMR spectroscopy in order to evaluate the hypothesis of grade-independent association between AK and SCC. The association between AKs and SCCs has also been evaluated using histopathology approaches. 

## 2. Results

### 2.1. Metabolomic Results

All the spectra obtained from the 35 tissue samples from AK/SCC patients and the 10 controls were considered for the analyses. We identified more than 40 metabolites in the region between 7.50 and 0.70 ppm of each spectrum, and all the assignments are listed in Appendix A.

The representative 1D ^1^H HR-MAS Carr–Purcell–Meiboom–Gill (CPMG) spectra of the five groups are reported in Figure 1, where the major signals of metabolites are labelled. The spectral region richest in metabolites lies in the range of 4.8–0.7 ppm, where the following molecules were identified: taurine (Tau), aspartate (Asp), serine (Ser), acetate (Ac), N-acetyl (N-Ac), β-glucose (β-Glc), glycine (Gly), valine (Val), lactate (Lac), alanine (Ala), myo-inositol (Myo), ascorbate (Asc), glutathione (GSH), scyllo-inositol (Scy), pyroglutamic acid (PGA), glutamine (Gln), glutamate (Glu), creatine (Cr), ethanolamine (EtA), choline (Cho) and ethanol (EtOH).

#### 2.1.1. PCA Analysis

Principal component analysis (PCA) of CPMG spectra showed spontaneous clustering for healthy controls, AK I, AK II, AK III and SCC, with the last two groups having more dispersed values in the scores plots (Figure 2a). In particular, negative PC1 values and positive PC2 values seem to be able to separate SCC and AK I from healthy controls, AK II and AK III. Negative PC2 values separate healthy controls and, at least partially, AK II, whereas AK IIIs are not characterized by any kind of PC value (Figure 2a). Analysis of the metabolic principal component loadings, revealed that healthy samples and AK II had the poorest content in metabolites among all groups (negative PC1 and PC2, Figure 2b). The scattering of the samples in the PC1 direction (PC1 positive in the scores plot, Figure 2a) is mainly due to a contribution of residual signals from lipids (PC1 loadings profile, Figure 2b). The PC2 positive loadings show the highest content in metabolites signals, characterizing especially SCC, AK I and AK III.

In order to obtain more detailed information on AKs and SCC potential biomarkers, lipids residues and glycerol signals were excluded from the analysis. This approach enabled better appreciation of clustering of metabolite expression within AKs and SCC groups with a new unsupervised PCA (Figure 3). As shown in Figure 3a, the scores plot highlights a clearer separation among the groups, with AK III being represented by negative PC3 values, whereas AK I is clustered together with SCC samples by PC3 positive values. These separations are due to groups of metabolites that are underlined in the loading profile in Figure 3b. On the other hand, negative PC1 scores plot partially cluster healthy controls and AK II.

Observing the PC3 loading profile of metabolites (Figure 3b), the AK III samples turned out to be the richest in PGA, threonine (Thr), Ser and two aromatic metabolites, phenylalanine (Phe) and tyrosine (Tyr), with respect to other AKs and SCC, while being the poorest in Lac and Myo. Taken together, our qualitative data suggest that the metabolic profile of AK I is more similar to that of SCC than to those of healthy skin, and also AK II and III.

#### 2.1.2. Quantitative Analysis

Loading profile from PCA cannot discriminate among groups based on any single metabolite, but rather considering a pool of metabolites. For this reason, the metabolite signals, highlighted by the PCA analysis, were deconvoluted to quantify the amount of metabolites and to better understand which metabolic pathways are involved in this association. 

Figure 4 reports the quantification of those metabolites that vary among the classes. Healthy tissue was confirmed to have a reduced metabolite content with respect to AK and SCC specimens. Indeed, compared with healthy controls, AK and SCC samples displayed increased content in every analyzed metabolite, with the exception of Glc and Scy that only show minimal variations. Not only SCCs, but also AK I, AK II and AK III turned out to be characterized by increased amounts of small metabolites, such as Lac, Ser, Gly, glycerophosphocholine (GPC), Cho, Glu, methionine (Met), Ala, GSH, phosphocholine (PC), EtA, Gln and Val. The content of Myo is increased only in SCC, while AK II and healthy controls display the same amount of Thr and PGA. T-test comparison of healthy tissue versus all the other groups identified significant changes in the content of many of the evaluated metabolites: Val, Ala, Met, Glu, Gln, EtA, PC, GPC, Tau, Lac and Asc (Appendix A).

#### 2.1.3. Cluster Heat Map

We elaborated a cluster heat map to investigate associations among healthy controls, AKs and SCC in term of specific metabolites (Figure 5). It appears that AK I, SCC and AK III display a more similar metabolic content compared with healthy controls and AK II. In more detail, three major clusters of metabolites that associate AK I, SCC and AK III were shown. Cluster 1 comprises the low levels of PGA, Ser and Gly displayed by SCC and AK I, whereas higher levels of these metabolites were reported in AK III. Likewise, Cluster 2 shows similar amounts of PC, GSH and Met in AK I and SCC, compared with lower contents of these metabolites in AK III. Cluster 3 shows the highest levels of Val, Gln and GPC in SCC and AK III with respect to AK I. We can also observe that healthy control and AK II have several metabolites in common.

#### 2.1.4. ANOVA Analysis

ANOVA analysis was performed on all the deconvoluted metabolites. Glu, Gln and Met are able to significantly discriminate AK and SCC specimens from healthy controls (Table 1). Ser, Gly and PGA differentiate AK III from the SCC specimens. Furthermore, SCC samples display a similar amount of these metabolites as in AK I, confirming the results of the cluster analysis (Figure 5, cluster 1). At the same time, AK I and AK II displayed lower levels of Glu, Myo and EtA compared to SCC. Cho and Myo are more expressed in SCC and discriminate this class from healthy control and AKs (Table 1).

### 2.2. Histopathological Results

A total of 170 cases of AK with or without SCC from newly diagnosed patients were selected for further histopathological evaluation (Table 2). Patients’ mean age was 75.4 ± 6.2 years. The majority of patients were male (72.4%) and most lesions were located on head and neck area (82.9%).

In total, 63 (37%) AK I, 77 (45%) AK II and 30 AK III (18%) were analyzed. Among these, 45 (26%) AK were associated with SCC, either in situ or invasive. Head and neck region was the most frequent site in which AK was associated with SCC (36/45, 80.0%), followed by upper limbs (7/45, 15.5%). Two patients (4.4%) were affected by chronic myeloid leukemia. SCC was more frequently associated with AK I (20/63, 31.7%) compared with AK II (18/77, 23.4%) and AK III (7/30, 23.3%) (*p* < 0.001). Considering only the 45 specimens displaying both SCC and AK, 44.4% (20/45) of SCC were reported in association with AK I, 40.0% (18/45) were associated with AK II and only 15.6% (7/45) were associated with AK III. 

Analysis of histopathology of specimens also showed that SCCs associated with AK I had a significantly higher Breslow thickness (see Appendix A) in comparison with those associated with AK II or AK III (*p* = 0.005, Table 2). Parakeratosis was less frequently reported in AK I specimens (37/63, 58.7%) than in AK II (68/77, 88.3%) or AK III (29/30, 96.7%) samples (*p* < 0.001). Overall, parakeratosis was more frequent in AK samples without SCC (107/125, 85.6%) than in those associated with SCC (27/45, 60.0%; *p* < 0.001). Similarly, hyperplasia was less frequent in AK I specimens (12/63, 19.0%) than in AK II (43/77, 55.8%) or AK III (23/30, 76.7%) samples (*p* < 0.001). No differences in the presence of hypertrophy were reported across AK I, AK II and AK III groups (Table 2). 

In the multivariate logistic-regression model the presence of parakeratosis in AKs appeared to be less frequently associated with SCCs (*p* < 0.001), while AKs with hypertrophy presented a more than two-fold higher risk of being associated with SCC (*p* < 0.05) (Table 3).

## 3. Discussion

Historically, the progression from AK to SCC was proposed to be dependent on the grade of AK, occurring according to a multi-step pathway [9]. However, this model has been extensively debated [20,21], and recent histological data suggest that all AK lesions can be associated with SCC, regardless of the thickness of epidermal changes [11,22]. The metabolic and histological features of AK associated with SCC remain to be fully investigated. Therefore, we conducted a comprehensive analysis to identify those features. We used histopathology and NMR; this latter approach has already contributed greatly to cancer research, unveiling complex biological mechanisms and deregulated pathways [23,24,25].

Although with all the limitations of any metabolic/histological study conducted on a limited number of specimens, we have shown: (i) Increased metabolism in AK and SCC as compared with healthy skin and different metabolic profiles among AK I, AKII, AKIII and SCC; and (ii) histological association between AK I and SCC. These findings are discussed below.

### 3.1. Increased Metabolism in AK and SCC as Compared with Healthy Skin and Different Metabolic Profile among AKI, AKII, AKIII and SCC

PCA shows a spontaneous clustering of all the different AK grades and SCC in terms of complete metabolic profile of the tissues, while healthy controls display a different profile (Figure 2a). 

The analysis of loading profiles show that healthy controls are poor in metabolites compared with the pathological groups. At the second PCA analysis, the lipid content was removed to allow an improved clustering of the groups. From the loadings profile, we identified a pool of metabolites, such as PGA, Thr, Ser, Phe, Tyr and also Lac and Myo that discriminate the AK III from the other pathological groups (Figure 3b). 

Our study shows a qualitative variation in the metabolic profile among skin samples of patients with AK, SCC and healthy subjects. In line with the results of the metabolomic qualitative analysis, the quantitative analysis confirms that samples from healthy skin have the poorest metabolite content when compared with specimens of AK and SCC (Figure 4). In particular, not only SCCs or AK II and III but also AK I were characterized by significantly increased amounts of small metabolites (Lac, Ser, Gly, GPC, Cho, Glu, Met, Ala, GSH, Thr, PC, EtA, Gln, PGA and Val). Among these metabolites, Cho is worth particular attention because this molecule plays a role in the synthesis of membrane phospholipids and its role in tumorigenesis has already been shown. Indeed, evidence suggests that phosphatidylcholine, as well as choline metabolites derived from its synthesis and catabolism, contribute to both proliferative growth and programmed cell death [26,27].

The imbalance of GSH metabolism may be associated with major alterations of the redox state of the pathological tissue that could increase the requirement of antioxidant molecules, such as GSH, and it is a metabolomics signature of sun-exposed skin tissues [28]. This suggests an increase of skin metabolism both in SCC and in AK lesions of all grades, in line with our previous findings [19].

The heat map clustering analysis (Figure 5) showed that six metabolites (PGA, Ser, Gly, PC, GSH and Met) are similarly expressed in AK I and SCC, with a different expression in AK III. Val, Gln and GPC are highly expressed in AK III and SCC, and the same trend is also reported in AK I. These results define three different clusters of expression among AK I, AK III and SSC. On the other hand, AK II, instead of clustering between AK I and AK III as could be expected, shows some similarities with healthy controls. Much of the data collected reaches the threshold of significance, as demonstrated by ANOVA analysis. 

ANOVA analysis shows that, few metabolites tend to discriminate AK I from SCC, while some metabolites (Ser, PGA, Gly, Thr, GSH) separate SCC from AK III. Furthermore, AKII showed similarities with healthy controls, thus confirming the findings observed during the heat-map analysis. Three metabolites, Glu, Gln and Met, were significantly increased in AK and SCC specimens compared with healthy controls. Glu is one of the major reservoirs of nitrogen in cells, and it is used as nitrogen donor for synthesizing many nitrogen compounds in both pre-cancer and cancer cells [29]. Gln is an essential bioenergetic and anabolic substrate for many cancer types, since cancer cells exhibiting aerobic glycolysis (Warburg effect) also rely on Gln for energetic metabolism [30]. Met is crucial for the growth and metabolism of cancer cells [31].

The levels of Ser, Gly and PGA observed in AK I specimens were comparable to those reported in SCC, while only lower levels of Glu and EtA distinguished low-grade AKs from SCC. Ser has a central role in supporting tumor growth and contributes to the anabolic pathways involved in the synthesis of glutathione, nucleotides, phospholipids and other metabolites [32]. PGA is an intermediate metabolite involved in the GSH pathways [33]. The increase of GSH metabolism reflects a high oxidative stress and is known to be a metabolomic signature of sun-exposed skin tissues [28]. These findings suggests that low-grade AKs show metabolic alterations similar to those reported in SCC. However, low levels of EtA distinguished low-grade AKs from SCC. Mammals cannot synthesize EtA, and thus this molecule is obtained from the diet as free EtA or in the form of phosphatidylethanolamine (PE), which is degraded by phosphodiesterases to yield glycerol and EtA. It has been shown that PE accumulates on the surface of some cancer cells. We can therefore speculate that higher levels of EtA are present in SCC as compared with AK I, thus reflecting enhanced metabolism of cancer cells.

On the other hand, AK III samples were richest in PGA, Ser, Phe, Tyr and Thr compared with AK I, AK II and SCC, while being the poorest in Lac and Myo. In particular, synthesis of Lac from glucose/glutamine represents one of the consequences of altered metabolism in cancer cells. Lac, in the tumor microenvironment, has a role in promoting tumor-cell proliferation and growth [34]. It is difficult to define a difference between AK III and low-grade AKs in terms of metabolic profile, but the poorest content in Lac can suggest a lower activation of neoplastic biochemical pathways.

In short, with all the limitations of any metabolic study on a limited number of specimens, our analysis seems to also suggest that the metabolic profile of AK I may be considered more similar to that of SCC than to those of healthy skin controls.

### 3.2. Histological Association between AK I and SCC

Histopathological analysis, conducted on a larger number of specimens of any-grade AK either associated, or not, with SCC, showed that AK I is more frequently associated with SCC than AK II and AK III. This finding supports some data showing that SCC can arise directly in an epidermis where the alterations are limited to the inner layers [11]. Furthermore, our data show the lack of correlation between dysplasia severity and the clinical, histological, thickening of the stratum corneum. These data may support the hypothesis that AKs, independently from their clinical appearance, can behave as an invasive SCC, as already suggested [12,35,36]. Since we analyzed only patients with a new diagnosis of AK, either already associated, or not, with SCC, we are not able to retrieve any consideration about the potential risk of progression from AK to SCC and its time frame.

Last, multivariate analysis showed that AKs displaying parakeratosis are less frequently associated with SCCs, while AKs with hypertrophy have a more than two-fold higher risk of being associated with SCC. Parakeratosis is defined as the presence of nucleated keratinocytes in the stratum corneum, and it is likely due to accelerated keratinocyte turnover. SCCs is an aggressive malignancy which however, in an apparent paradox, retains a squamous differentiated phenotype except for very dysplastic lesions [37]. Gandarillas et al. proposed a model in which squamous differentiation response is triggered by DNA-damage (DNA damage-differentiation response (DDDR)) and removes precancerous, genetically altered cells. Due to this self-defense mechanism, cell cycle deregulation caused by oncogenic alterations first results in proliferation, but shortly triggers the mitosis checkpoints and squamous differentiation. We speculate that the presence of parakeratosis can be a hallmark of an altered, but still present, pathway of squamous differentiation that can contribute towards removing dysplastic cells from the skin [38].

Furthermore, the model proposed by Gandarillas et al. [38] contemplates that progressive loss of the DDDR would allow damaged cells to proliferate, leading to overt SCC. In our study, hypertrophy of AK, defined as a proliferation of cells in the epithelium, seems to increase the risk of association with SCC.

## 4. Material and Methods

### 4.1. Study Setting and Design

This study was conducted at the Department of Dermatology of Modena University Hospital (Italy) from May 2016 to December 2020. The study was conducted according to the Helsinki Declaration, after approval of the study design by the local Ethical Committee (pratica 8/16, Comitato Etico dell’Area Vasta Emilia Nord, Policlinico di Modena, Via Largo del Pozzo 71, 41124). All patients signed an informed consent form before inclusion in the study.

Patients were analyzed by either metabolomics, histopathology or both approaches (see below for details). AK grade and SCC histology were defined according to [9,39,40].

### 4.2. Sample Storage and Preparation

All skin samples were collected by punch biopsy of a 6-mm diameter and sectioned into two parts. One part underwent routine histological assessment while the other part was dissected to separate skin from adipose tissue, directly frozen in liquid nitrogen and stored at −80 °C until metabolomic analysis. In the healthy subjects, biopsies were collected from normal skin adjacent to the pigmented benign lesion.

### 4.3. Metabolomic Study

#### 4.3.1. Patients

In total, 45 consecutive adult subjects were enrolled: 35 patients newly diagnosed with AK (AK I, *n* = 7; AK II, *n* = 8; AK III, *n* = 9) or SCC (*n* = 11), undergoing biopsy for histological confirmation, and 10 healthy subjects undergoing preemptive routine skin surgery for pigmented benign lesion removal for aesthetic reasons (e.g., dermal nevi).

#### 4.3.2. HR-MAS NMR Measurements

Samples (20–30 mg, from biopsy or surgical resection) were analyzed through ex vivo HR-MAS NMR spectroscopy, without any pretreatment. All sample were weighed and then introduced into the 50 µL MAS zirconia rotor (4 mm OD) with 10 µL of deuterated water (D_2_O) closed with a cylindrical insert to increase sample homogeneity and with the appropriate cap. ^1^H and ^13^C HR-MAS spectra were recorded with a Bruker Avance III HD 600 MHz spectrometer, operating at 600.13 and 150.90 MHz, respectively. The HR-MAS probe temperature was controlled by a Bruker cooling unit. Experiments were performed at a temperature of 5 °C to prevent tissue degradation processes [41]. The set up took about 20 min. We performed one-(^1^H or ^13^C) and two-dimensional experiments (1D and 2D, respectively).

The three different types of 1D proton spectra were acquired using sequences implemented in the Bruker software: (i) a standard sequence with water presaturation during relaxation delay, (ii) a water-suppressed spin-echo CPMG (Carr–Purcell–Meiboom–Gill) sequence with water presaturation during the relaxation delay and (iii) a sequence for diffusion measurements based on stimulated echo and bipolar-gradient pulses (LED). In addition, 2D homonuclear and heteronuclear techniques (COSY, TOCSY and HSQC) were used for the metabolomics characterization. For the experimentals details, see Righi et al., 2019 [42].

#### 4.3.3. Data Processing 

The NMR spectra were preprocessed using the TOPSPIN software package (Version 3.5, Bruker Biospin, Rheinstetten, Germany). The ^1^H CPMG HR-MAS NMR spectra obtained for each tissue sample were subjected to 0.5 Hz exponential line broadening and zero-filled to 64 k points prior to Fourier transformation and then phased and baseline corrected manually.

After this preliminary processing, the spectra were prepared for chemometric analysis using the MNova software package (MestReNova, ver. 11. 0, 4-18998, 2017 Mestrelab Research S. L., Santiago de Compostela, Spain).

The spectral regions δ 4.8–0.5 ppm were normalized with respect to sample weight, then aligned and binned (δ 0.01 ppm). Deconvoluted areas of selected peaks from ^1^H CPMG HR-MAS spectra were obtained with an automated/controlled fitting routine based on the Levenberg–Marquardt algorithm applied after manual peak selection, adjusting peak positions, intensities, line widths and Lorentzian/Gaussian ratios, until the residual spectrum was minimized [43].

#### 4.3.4. Statistical Analysis

Whole spectral profiles were analyzed, after pareto scaling, through multivariate statistical analysis. In particular, principal component analysis (PCA), an unsupervised multivariate method, was performed to examine the intrinsic variation in the data set. We excluded the regions with residual fatty acids and mepivacaine signals from the analysis. The data were visualized in the form of the PCA score plots and loading plots. Each data point on the score plots represented an individual sample, and each data point on the loading plots represent one NMR spectral data point related to the metabolites.

The metabolite signals highlighted by multivariate analysis were deconvoluted and the relative concentrations of 25 specific metabolites were estimated by using the areas of selected peaks spectra. Data were reported as means ± standard errors (arbitrary units). For Student’s *t*-test, a paired two-sample test was used to determine the means. A *p*-value < 0.05 was considered statistically significant. One-way ANOVA, PCA and PLSDA analyses were performed. The heat maps were generated using the differential biomarkers of importance identified among the five groups. Post hoc test (pair-wise multiple comparisons) was also used to determine the significant pair(s) after ANOVA. 

All qualitative and quantitative analyses were performed by using the MetaboAnalyst 5.0 program, a free web-based metabolomics data analysis software [44].

### 4.4. Histopathology Study

The cases selected for the histopathological analysis included the 24 consecutive AK pathological specimens that underwent metabolomic examination and 146 consecutive other specimens of AKs from newly diagnosed patients, from the archive of the Anatomic Pathology Unit at the University of Modena and Reggio Emilia.

The following clinical parameters were assessed: gender, age, site (head and neck, trunk, limbs).

Skin excisional or incisional biopsies were fixed in 4% buffered formalin and embedded in paraffin. For AK lesions, histologic examination was performed on sections stained with hematoxylin and eosin, assessing the following features: grade (I, II or III), parakeratosis/orthokeratosis/hyperplasia/hyperthrophia/acantholyis/clear cells/elastosis/atrophy/pigmentation/ulceration (presence or absence) and inflammation (mild or moderate or severe).

Histological features evaluated for SCCs were grades of differentiation (G1—well differentiated, G2—moderately differentiated or G3—poorly differentiated), Breslow thickness (mm), Clark level (1—epidermis, 2—papillary dermis, 3—papillary-reticular dermis, 4—reticular dermis, 5—subcutaneous fat), perineural invasion, ulceration, metastasis, recurrence (presence or absence), concomitant AK and its features (the same evaluated for AKs that are not associated to SCC: grade, parakeratosis, orthokeratosis, acantholysis, clear cells, hyperplasia, elastosis, atrophy, hypertrophy, inflammation, pigmentation, and ulceration).

Statistical analysis was performed using STATA software version 14 (StataCorp, 2015, Stata Statistical Software: Release 14, College Station, TX: StataCorp LP). Descriptive statistics were presented for baseline demographic clinical characteristics for the entire group, as well as for the single groups of patients. Continuous variables were presented as the number of patients (n), mean, standard deviation (SD), minimum (min) and maximum (max). Comparisons between subgroups were performed using unpaired Student’s *t*-test for two groups or ANOVA for more than two groups; while categorical variables were presented as frequencies (N, percentage (%)) and compared using Pearson’s chi-squared test. The association between these parameters and AK with or without SCC status was assessed using a logistic regression model with stepwise forward selection. A multivariate logistic regression model was carried out using a stepwise selection method to identify the prognostic factors between groups. In the first step, the intercept-only model was fitted and individual score statistics for the potential variables were evaluated. A significance level of *p* < 0.05 was used to allow a variable into the model. In stepwise selection, an attempt was made to remove any insignificant variables from the model before adding a significant variable to the model. Hosmer and Lemeshow test was used to evaluate “goodness of fit” in the selection model. Data from the univariate and multivariate logistic regression analyses were expressed as odds ratio (OR) and 95% confidence interval (CI). A *p* < 0.05 was considered statistically significant.

## 5. Conclusions

Our findings, derived from metabolomics and histopathological data, support the notion that AK I are different from healthy skin and share different features with SCCs. Indeed, they are metabolically active lesions with metabolic profile similar to high-grade AKs and to SCC. Therefore, we speculate that AK I lesions also have the potential to evolve directly into SCC. Moreover, the negative association of AKs with parakeratosis and the positive association with hypertrophy suggest a similar behavior between AKs and SCCs. This may further support the expanding notion that all AKs should be treated independently from their clinical appearance or histological grade because they may be associated with SCC. This has an immediate clinical implication: all AKs should be treated independently from their clinical appearance or histological grade because it is not possible to predict their potential evolution to SCC.

## Figures and Tables

**Figure 1 cancers-13-05560-f001:**
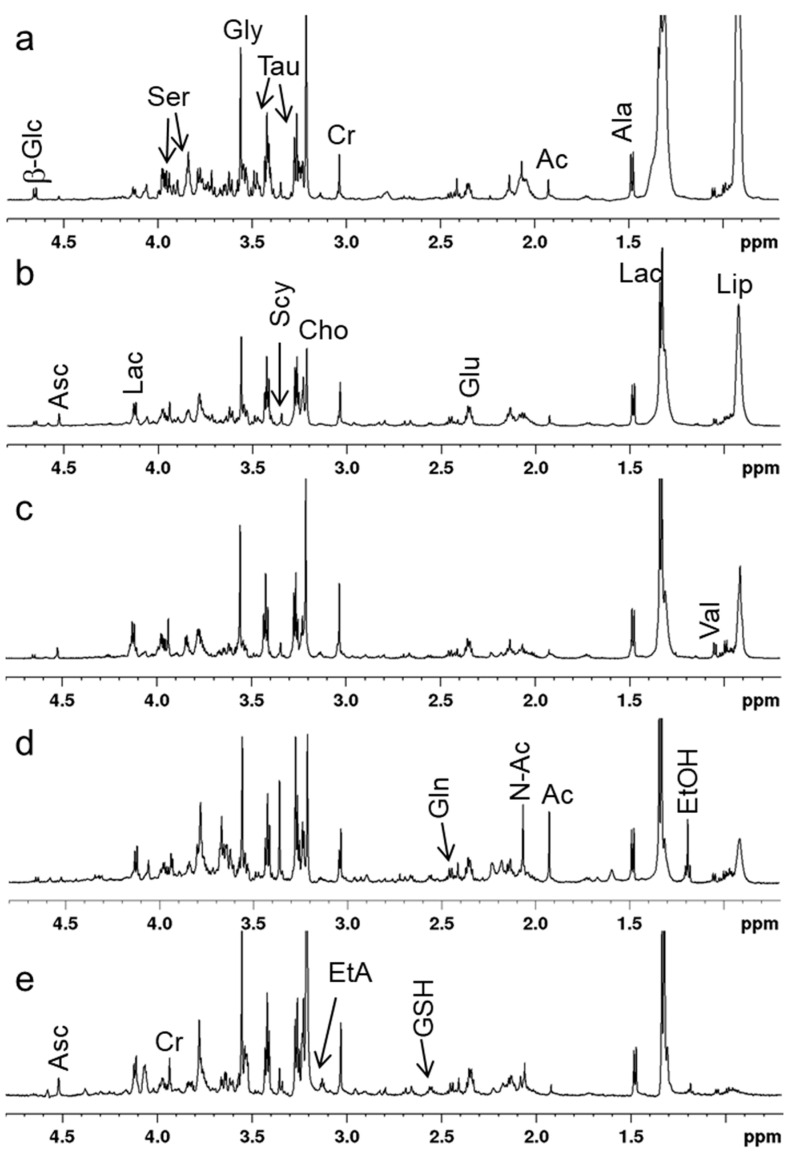
Representative 1D ^1^H Carr–Purcell–Meiboom–Gill (CPMG) high-resolution ^1^H magic angle spinning (HR-MAS) spectra between 4.8 and 0.7 ppm of healthy (**a**), AK I (**b**), AK II (**c**), AK III (**d**) and (**e**) SCC are reported. Major metabolites are labelled. Acetate (Ac), alanine (Ala), ascorbate (Asc), creatine (Cr), glutamate (Glu), β-glucose (β-Glc), lactate (Lac), lipids (Lip), N-acetyl (N-Ac), serine (Ser), taurine (Tau), valine (Val).

**Figure 2 cancers-13-05560-f002:**
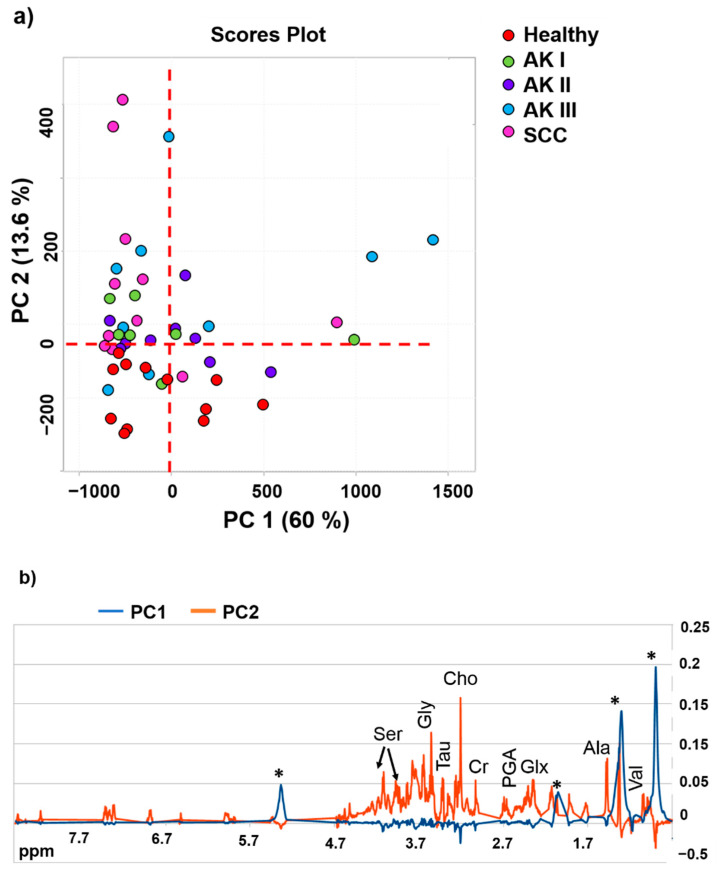
(**a**) PCA scores plot and (**b**) loadings profiles of PC1 and PC2 obtained from the spectral CPMG data of the five classes of samples. Legend: *—lipids residues, Val—valine, Ala—alanine, Glx—glutamate and glutamine, PGA—pyroglutamic acid, Cr—creatine, Cho—choline, Tau—taurine, Gly—glycine, Ser—serine.

**Figure 3 cancers-13-05560-f003:**
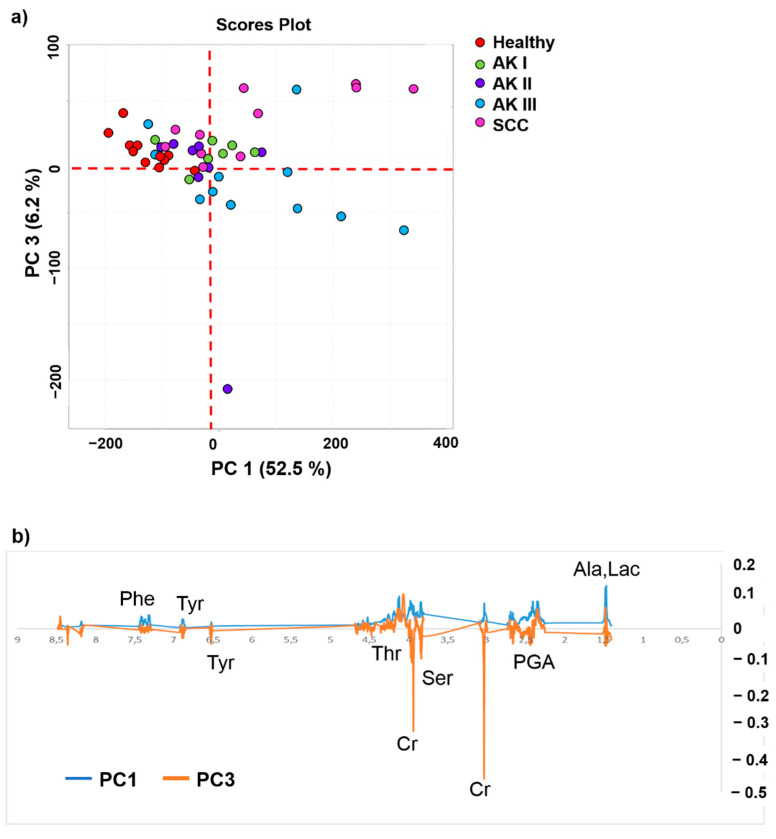
(**a**) PCA scores plot and (**b**) loadings profiles of PC1 and PC3 obtained from the spectral data of the five classes. In (**b**) labelled metabolites: Phe—phenylalanine, Tyr—tyrosine, Thr—threonine, Cr—creatine, Ser—serine, PGA—pyroglutamic acids, Ala—alanine, and Lac—lactate.

**Figure 4 cancers-13-05560-f004:**
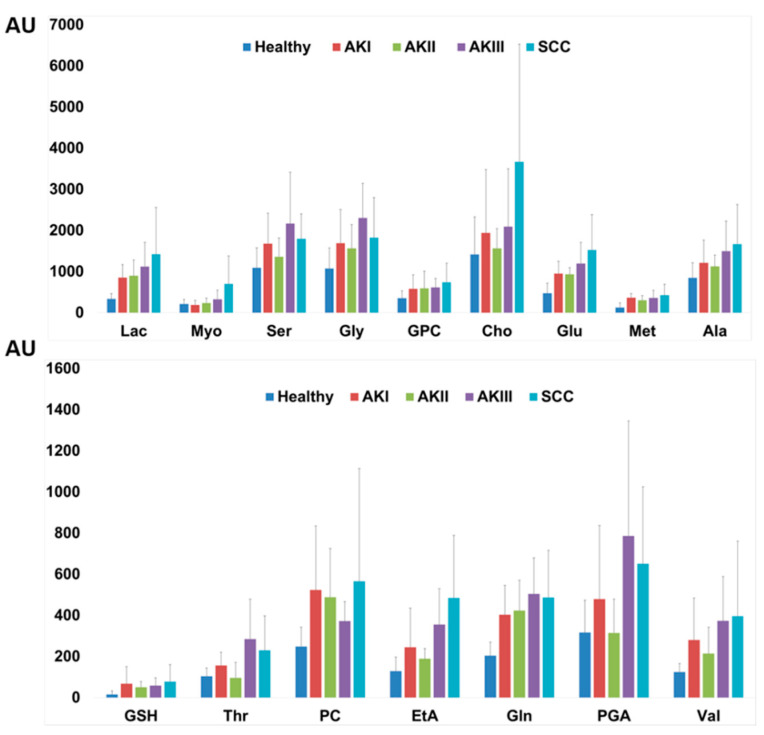
Selected small metabolites highlighted by qualitative multivariate analysis were deconvoluted and quantified from ^1^H CPMG spectra and the mean ± SD are reported. The following metabolites are reported on the X axes: lactate (Lac), myo-inositol (Myo), serine (Ser), glycine (Gly), glycerophosphocholine (GPC), choline (Cho), glutamate (Glu), methionine (Met), alanine (Ala), glutathione (GSH), threonine (Thr), phosphocholine (PC), ethanolamine (EtA), glutamine (Gln), pyroglutamic acid (PGA), valine (Val). Y axes—arbitrary units.

**Figure 5 cancers-13-05560-f005:**
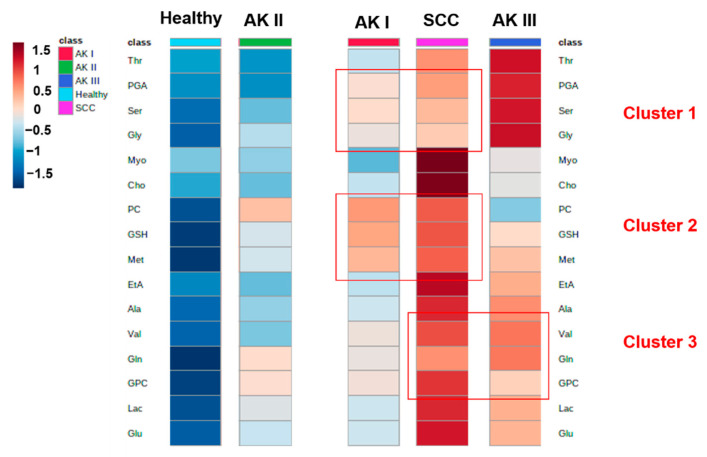
Cluster heat map with all 5 classes are reported. Three major metabolites clusters are identified: Cluster 1: PGA, Ser and Gly; Cluster 2: PC, GSH and Met; and Cluster 3: Val, Gln and GPC.

**Table 1 cancers-13-05560-t001:** The one-way ANOVA results and post hoc test are reported. Healthy control (0), AK I (1), AK II (2), AK III (3) and SCC (4).

	f-Value	*p*-Value	log10 (*p*-Value)	FDR	Fisher’s LSD
**Glutamate**	5.6949	0.001004	2.9983	0.015728	1-0; 2-0; 3-0; 4-0; 4-1; 4-2
**Ethanolamine**	5.5884	0.001138	2.9438	0.015728	3-0; 4-0; 4-1; 3-2; 4-2
**Glutamine**	5.3167	0.001573	2.8033	0.015728	1-0; 2-0; 3-0; 4-0
**Glutathione**	4.9945	0.002321	2.6343	0.017409	1-0; 3-0; 4-0; 2-1; 4-2; 3-1; 3-4
**Methionine**	4.2611	0.005762	2.2395	0.034569	1-0; 2-0; 3-0; 4-0; 4-2
**Pyroglutamic acid**	4.0771	0.007273	2.1383	0.036367	3-0; 4-0; 3-1; 3-2; 3-4; 4-2
**Lactate**	3.7253	0.011419	1.9424	0.045287	3-0; 4-0
**Threonine**	3.6821	0.012076	1.9181	0.045287	3-0; 4-0;3-1; 3-2; 4-2; 4-3
**Myo-inositol**	3.3984	0.017471	1.7577	0.058237	4-0; 4-1; 4-2; 4-3
**Glycine**	3.1441	0.024413	1.6124	0.067903	3-0; 4-0; 3-2; 3-4
**Taurine**	2.97	0.030749	1.5122	0.076389	1-0; 2-0; 3-0; 4-0
**Choline**	2.7793	0.039645	1.4018	0.076389	4-0; 4-1; 4-2; 4-3
**Serine**	2.7589	0.040741	1.39	0.076389	3-0; 4-0; 3-1 3-2; 3-4; 4-2

**Table 2 cancers-13-05560-t002:** Descriptive analysis of the AK specimens analyzed.

	AK I (*n* = 63)		AK II (*n* = 77)		AK III (*n* = 30)		Total (*n* = 170)		*p*-Value
	N	%	N	%	N	%	N	%	
**Gender**									
Female	16	25.4	25	32.5	5	16.7	46	27.1	0.220
Male	47	74.6	51	66.2	25	83.3	123	72.4	
**Site**							0		
Limbs	13	20.6	11	14.3	2	6.7	26	15.3	0.076
Head and neck	49	77.8	66	85.7	26	86.7	141	82.9	
Trunk	1	1.6	0	0.0	2	6.7	3	1.8	
**Associated SCC**	20	31.7	18	23.4	7	23.3	45	26.5	<0.001
**SCC grade**									
G1	3	15.0	4	22.2	3	42.9	10	22.2	0.064
G2	15	75.0	6	33.3	3	42.9	24	53.3	
G3	1	5.0	2	11.1	1	14.3	4	8.9	
in situ	1	5.0	6	33.3	0	0.0	7	15.6	
**SCC characteristics**									
**Breslow thickness, mm (median (IQR))**	1.8 (0.3–5)		0.5 (0.3–1.2)		0.6 (0.4–0.8)		0.9 (0.5–3.5)		0.005
**Clark level for SCC**									
2	4	21.1	9	75.0	5	71.4	18	47.4	0.062
3	5	26.3	1	8.3	1	14.3	7	18.4	
4	4	21.1	0	0.0	1	14.3	5	13.2	
5	6	31.6	2	16.7	0	0.0	8	21.1	
**Perineural invasion**	0	0.0	1	6.7	0	0.0	1	2.4	0.411
**Ulceration**	6	31.6	4	57.1	3	42.9	13	31.7	0.749
**Metastasis**	1	7.1	0	0.0	0	0.0	1	3.2	0.534
**Recurrence**	1	7.1	1	9.1	0	0.0	2	6.5	0.759
**AK characteristics**									
**Parakeratosis**	37	58.7	68	88.3	29	96.7	134	78.8	<0.001
**Orthokeratosis**	16	25.4	34	44.2	12	40.0	62	36.5	0.065
**Acantholysis**	3	4.8	15	19.5	13	43.3	31	18.2	<0.001
**Clear cells**	24	38.1	18	23.4	9	30.0	51	30.0	0.167
**Hyperplasia**	12	19.0	43	55.8	23	76.7	78	45.9	<0.001
**Elastosis**	63	100	77	100	29	96.7	169	99.4	0.096
**Atrophy**	29	46.0	35	45.5	12	40.0	76	44.7	0.847
**Hypertrophy, grade**									
0	38	60.3	34	44.2	12	40.0	84	49.4	0.203
1	25	39.7	42	54.5	18	60.0	85	50.0	
2	0	0.0	1	1.3	0	0.0	1	0.6	
**Inflammation, grade**									
0	16	25.4	7	9.1	1	3.3	24	14.1	0.029
1	27	42.9	32	41.6	11	36.7	70	41.2	
2	14	22.2	26	33.8	12	40.0	52	30.6	
3	6	9.5	12	15.6	6	20.0	24	14.1	
**Pigmentation**	17	27.0	15	19.5	5	16.7	37	21.8	0.427
**Ulceration**	5	7.9	1	1.3	5	16.7	11	6.5	0.034

**Table 3 cancers-13-05560-t003:** Univariate and multivariate logistic regression model of the risk of association between histopathological features of AK and SCC.

	Univariate			Multivariate		
	OR	95% CI	*p*-Value	OR	95% CI	*p*-Value
**Parakeratosis**	0.25	(0.11–0.54)	0.001	0.19	(0.08–0.46)	**<0.001**
**Orthokeratosis**	0.74	(0.35–1.49)	0.385			
**Acantholysis**	0.47	(0.17–1.32)	0.156			
**Clear cells**	0.68	(0.31–1.49)	0.344			
**Hyperplastic AK**	0.72	(0.36–1.44)	0.357			
**Elastosis**	1	empty				
**Atrophy**	0.52	(0.25–1.06)	0.076			
**Hypertrophy**						
**Absent**	ref.		0.045	ref.		
**Present**	2.01	(1.01–3.97)		2.75	(1.28–5.91)	**0.009**
**Inflammation, grade**						
**0**	ref.					
**1**	0.69	(0.25–1.88)	0.473			
**2**	0.66	(0.23–1.91)	0.452			
**3**	0.66	(0.19–2.33)	0.526			
**Pigmentation**	0.86	(0.37–2.01)	0.738			
**Ulceration**	0.24	(0.02–1.93)	0.181			

## Data Availability

Data sharing is not applicable to this article.

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
