# Peer review of "Metabolomic Analysis of Actinic Keratosis and SCC Suggests a Grade-Independent Model of Squamous Cancerization"

_cancers, 2021, doi:10.3390/cancers13215560_

Round 1

Reviewer 1 Report

I reviewed this interesting manuscript, which is better than the previous version. The authors responded all my comments and suggestions and now is more understandable. The supplement material is interesting, well explained and extensive. I do not have further comments to add. 

Reviewer 2 Report

I believe that the authors have considerably increased the quality of the text and significantly nuanced their statements. I believe that under these conditions, the manuscript deserves to be published in this journal.

Author Response

Dear reviewer,

Thank you for your work in reviewing the MS and for the positive comments.

This manuscript is a resubmission of an earlier submission. The following is a list of the peer review reports and author responses from that submission.

Round 1

Reviewer 1 Report

The manuscript is interesting and well redacted. The authors evaluated several metabolites related to AK and SCC, and they aimed to relate the metabolomic features between AK and SCC, providing interesting results. I only have minor comments that authors should take into account. 

Could it be possible to evaluate how many SCCs had tumor necrosis? 

Could it be possible to evaluate how many SCCs had tumor budding and also evaluate tumor stroma ratio? These features are important in tumor aggressiveness, and could be interesting to establish how many and which types of AK were related with this feature.

The authors evaluated important histopathologic cancer features (perineural invasion and ulceration), could it be possible to indicate which metabolite was related with these features?  

Would it be possible to use photomicrographs (Figure 1, 2...) showing the features described in tables?

The discussion is interesting and well redacted, the authors provide important information that may be used in further research; however, I consider that it could be interesting to describe the function of metabolites reviewed in the cancer development and their relationship with levels of AK. For example: "serine and glycine are related to cancer cell growth and metabolism. It was found a greater concentration of serine/glycine in SCC with AK (types 1, 2 or 3) also similar to serine/glycine. Choline, that is related to carcinogenesis and tumoral progression, was found in SCC with AK (types 1, 2 or 3).

Reviewer 2 Report

Comments to Rhigi et al.

In this paper, Rhigi et al. study a part of the metabolic profile of the different degrees of actinic keratosis (AK) and squamous skin carcinoma (SCC). The authors claim that they found metabolic similarities that might justify the evolution of AK I to SCC more frequently than other grades of AK. They also say they found histopathological data that support that result.

The work, in part, builds on previous studies that showed that all AK, regardless of grade, can evolve into SCC; therefore, all SCC should be treated given its potential malignancy. It is also partly based on the conclusions of a previous study (REF. 11 in the paper) where it was concluded that AK I would be the most frequent grade of AK associated with SCC and, therefore, would be the form of AK most able to transform into SCC. That finding would also be confirmed in this study by Rhigi et al.

The work is interesting, but we think it has several significant weaknesses:

-We think that the fact that AK I is the one that most frequently appears next to SCC would not permit the authors to infer that it is the degree of AK with the most significant potential to evolve to malignancy. Therefore, we do not think it is the right way to determine which grade of AK has the most malignant potential. To know that, it would be necessary to carry out a prospective study and show the percentage of each one of the AK grades that evolve to SCC. Authors can discuss this point and try to rely on the literature for it.

Of course, to take a series of SCCs and look at which grade of AK is the most often associated with SCC can give the result of AK I and, since it is known that it can evolve to malignancy, it would be its potential origin. However, this may be because AK I would be the most prevalent form, but not because its transformation rate to SCC would be higher.

This fact does not exclude this and other studies' essential conclusion that AK I must be treated because it can transform to SCC. Indeed, in absolute terms, if AK I is the most prevalent and since it can evolve to malignancy, it may be the most frequent cause of malignancy to SCC. However, the AK I rate of transformation does not have to be the highest (we could say the "most premalignant grade"). We believe that this critical difference is not demonstrated in the paper and should be discussed by adding literature. Thus, we also believe that it is unlikely for the AK I to be the closest to SCC in biological terms, including in the metabolic area.

-As for the metabolomic differences identified. The study identifies just over 40 differential metabolites, of which some distinguish between AK and SCC, and there is limited N of study, something that the authors correctly indicate in the discussion. Thus, the assertion of differences in the metabolic footprint between these entities may be premature and should be limited to indicating that differences have been found in several specific metabolites and not in the metabolomic fingerprint in the broad sense, which in our opinion, would be a more ambitious study.

-Concerning the differences in metabolites between the entities studied, the separation of the clusters presented in Figures 1 and 2 is not evident. The authors may also present a dendrogram showing unsupervised clustering; something they can do from the values derived from the study of principal components; it can be done with all or part of the metabolites differentially found.  

-The authors indicate that the metabolomics between AK I and SCC is similar. Here, also, we have several criticisms: (i) the similar levels of the indicated three metabolites would not allow the authors to conclude that these two entities are metabolically similar, in the broad sense, but that they would be similar in the levels of those three metabolites. (ii) In Figure 4, it does not seem very evident that the levels of these three metabolites are similar between AK I and SCC. (iii) From this possible similarity in these three metabolites between AK I and SCC, it cannot be concluded that AK I is more dangerous than AK grades II and, above all, III in their capability to evolve to SCC.

-In the study of histopathology, it is not clear to us that the facts that AK-I has a negative association with parakeratosis and a positive one with hypertrophy define the degree of AK I as the one with the most significant potential to evolve to SCC.

-Regarding Table 2, it may be sufficient to show the P-value of the ANOVA, the P-value of the post hoc test between each comparison, and indicate the significant comparisons.

Round 2

Reviewer 2 Report

Comments to Rhigi et al. (second revision)

 The authors have improved the work; however, I think that the main criticisms I made remain. Thus, the biggest criticisms I would make would be: (i) the work does not evaluate the hypothesis of "grade-independent progression from AK to SCC, as stated in several parts of the paper. (ii) The metabolic study is not exhaustive to affirm that the metabolomic profile of these lesions is made. (iii) Finding minor similarities, for example, between AK I and SCC in three metabolites does not allow the authors to claim that they are similar in metabolomics. There must be many more differences in a more comprehensive study, as expected to exist at the genomic, transcriptomic, and epigenomic levels.

Some concrete examples would be the following:

-On line 18, I do not think that what has been done can be called "the complete metabolic fingerprint."

-Lines 19, 20

"... to evaluate the hypothesis of grade-independent progression from AK to SCC.

This paper is not evaluating what the authors claim. The same statement is made on lines 33, 34 and lines 89, 90.

-Lines 42, 43.

"... SCCs and AK I were found out to be significantly associated in terms of metabolite content."

There are only three similar metabolites, in terms of levels, between AK I and SCC. So the statement looks pretty ambitious to me.

-Lines 47, 48

..."This has an immediate clinical implication: all AKs should be treated independently from their clinical appearance or histological grade because it is not possible to predict their potential evolution to SCC."

This statement is not a conclusion that can be drawn from this study. It is not something that can be deduced from the resemblance in levels in very few metabolites. Although on the other hand, it is an increasingly accepted reality. In this study, it would be more of a point of discussion. However, it cannot be deduced from this study as a final implication.

-Figure 2C.

That is not the kind of dendrogram that was asked of the authors. The purpose was to demonstrate how skin lesions are grouped according to the different metabolic profiles to appreciate better where the groups of individuals (lesions) are, which we find confusing in the actual principal component diagrams.  

-Lines 136-138

"... Taken together, our data suggest that the metabolic profile of AK I and/or AK II is more similar to that of SCC than to those of healthy skin and also AK III."

 To make this statement, in the unsupervised cluster (dendrogram), these groups of lesions should be next to the SCC and further away from the AK III.

-Lines 158, 159

"... The levels of Ser, Gly and PGA observed in AK I specimens were similar to those reported in SCC specimens..."

Similar levels in AK I and SCC in only three metabolites do not justify being similar at the metabolic level as said in other parts of the manuscript (lines 263 and 264, for example).

- Lines 175 to 178.

Here, in the series of AK collected by the authors, the one that is most often associated with SCC is the AK I. Nevertheless, this does not mean that AKI has the most significant potential for malignancy; since it may be because AK I is the most frequent in the population. Thus,  to show which AK is the most transformed, the number of cases of each degree of AK collected must be proportional to that observed in the population. Alternatively, the authors can show references supporting that affirmation.

-Lines 263, 264.

"... Taken together, our data suggest that the metabolic profile of AK I and/or AK II is more similar to that of SCC than to those of healthy skin or AK III..."

I think the metabolic study done is not deep enough to make that claim; and that it is only valid for the few metabolites indicated.

-On line 16, it is repeated: "can progress to."